# Intensive Multimodal Chemotherapy in a Dog Suffering from Grade III/Stage IV Solid Mammary Carcinoma

**DOI:** 10.3390/ani14172618

**Published:** 2024-09-09

**Authors:** Claire Beaudu-Lange, Emmanuel Lange

**Affiliations:** Clinique Vétérinaire de la Pierre Bleue, 35550 Pipriac, France; lange.emmanuel@gmail.com

**Keywords:** solid canine mammary carcinoma, grade III, lymphatic intravascular invasion, stage IV, carboplatin chemotherapy, metronomic chemotherapy, toceranib, firocoxib, chloraminophene, QoL

## Abstract

**Simple Summary:**

Very few studies have proven chemotherapy efficacy against canine mammary carcinomas. As a result, we still lack efficient standardized protocols, which do exist in human cases, to treat these carcinomas or to adapt treatment according to the aggressiveness of the tumor in dogs. In this case report, we describe a highly aggressive mammary carcinoma; we applied an intensive treatment combining radical surgery and intensive adjuvant chemotherapy, which included metronomic and maximal-tolerated-dose chemotherapy, nonsteroidal anti-inflammatory drugs and inhibitors of tyrosine kinase. We obtained the informed consent of the owner, who was aware of the poor prognosis without treatment and of the fact that this combinatorial treatment had not been published yet. The treatment enabled this one dog to live for 218 days with very good quality of life. A large-scale prospective study will be needed to confirm this result.

**Abstract:**

Very few studies, often with very small cohorts, have proven chemotherapy efficacy against canine aggressive mammary carcinomas, either in terms of metastasis or median survival, in dogs after surgery and chemotherapy, with such outcomes not being confirmed by other studies. As a result, we lack efficient standardized protocols, which exist in human cases, according to the grade and stage of the tumor in dogs. In this case report, we describe a relapsing grade III solid mammary carcinoma evolving into prominent lymphatic intravascular invasion with multifocal nodal extension (stage IV); we applied an intensive treatment combining radical surgery and intensive adjuvant chemotherapy. The latter combined carboplatin maximal-tolerated-dose chemotherapy, with doses adjusted as necessary, and metronomic chemotherapy with firocoxib, toceranib and chloraminophene, progressively administered and carefully monitored. Adapting the doses prevented adverse events and resulted in 218 days of survival with good quality of life. To our knowledge, this is the first description of such a treatment combination. Our result should be confirmed with a large-scale prospective study.

## 1. Introduction

Very few studies have proven chemotherapy efficacy against canine aggressive mammary carcinoma. When used separately, some drugs had interesting effects on the metastases of mammary carcinoma [1] or on the median survival of small cohorts after surgery [2,3], but other studies failed to prove any effect of chemotherapy on median survival after surgery [4,5,6,7]. This lack of results is likely due the heterogeneity of the cohorts (type of surgery, histologic type/grade, accurate extension assessment and type of treatment) and the small number of patients included in most publications. As a result, we still lack standardized efficient protocols, which do exist in human cases, according to the histologic classification, grade and stage of the tumor in dogs [8]. In this case report, we describe a relapsing grade III solid mammary carcinoma evolving into prominent lymphatic intravascular invasion with multifocal LN metastatic extension (stage IV); we applied an intensive treatment combining radical surgery and intensive adjuvant chemotherapy. The latter combined carboplatin maximal-tolerated-dose chemotherapy, with doses adjusted as necessary, and metronomic chemotherapy with firocoxib, toceranib and chloraminophene, progressively administered and carefully monitored. Adapting the dose of each molecule prevented adverse events and resulted in 218 days of survival with good quality of life. To our knowledge, it is the first description of such a treatment combination. This treatment should be tested in a large-scale prospective study.

## 2. Materials and Methods

The doses of each molecule and the precise protocol are detailed in Table 1. The dog was hospitalized for 24 h for each chemotherapy treatment, as required by French law, and treated with intravenous maropitant (1 mg/kg/d) before each intravenous chemotherapy treatment at the maximum tolerated dose. Maropitant (1 mg/kg/d PO) was also systematically prescribed for 4 days following each carboplatin chemotherapy treatment.

Blood counts were performed on an Idexx Procyte Dx and biochemical analyses were performed on an Idexx Catalyst.

The adverse events were graded with VCOG CTCAE version 2 [9].

The scanner was a Philips Access (16-bar unit).

The fine-needle aspiration cytology was analyzed with May Grunwald Giemsa staining by Delphine Rivière, Inovievet laboratory.

The histological and immunohistochemical analyses were all performed at the Oniris laboratory by F. NGuyen, in the team led by J. Abadie, by using protocols already published elsewhere by their team. The immunohistochemical analyses were performed retrospectively. Briefly, 2–4 μm histoslides from formalin-fixed paraffin-embedded samples were subjected to hematoxylin–eosin–saffron (HES) staining for classical histology, using a Leika autostainer, and to automatized immunohistochemistry using a Benchmark XT automat (Roche diagnostics) according to the recommendations of the fabricant and routinely validated protocols. The primary antibodies used were against estrogen receptor (ER), progesterone receptor (PR), epidermal growth factor receptor type 2 (HER2), pancytokeratins (CK) and the cyclooxygenase type 2 enzyme (COX2).

Quality of life was systematically monitored in the form of an informal discussion with the owners at each visit, and any side effect or change in behavior was noted. The discussion concerning the quality of life was not structured in the form of a systematic written questionnaire, but we indicated all the points to be monitored (appetite; sleep; interactions with her owners and other animals; interest in her environment and usual games; the presence of signs of pain; difficulty walking, defecating, urinating, jumping, etc.; and the presence of side effects (gastrointestinal, fatigue, fever, edemas of the pelvic limbs, or any other unusual event)) by asking the owner to report them to us systematically. At each chemotherapy consultation, the owner was questioned on these very points.

## 3. Case Report

### 3.1. Clinical Presentation

A pinscher entire female dog, 8.5 years of age, weighing 4.0 kg, was referred for a relapse of a mammary tumor on 21 January 2021. She had had a unilateral right complete mastectomy 7 months prior at her general practitioner’s clinic, but the mammary tumor (size 3 cm) had not been subjected to a histopathological study. She had her season in the meantime, but the exact date is unknown. A mammary tumor relapse was observed on the scar one month prior to consultation and had grown very quickly since then. On the day of the referral consultation, the relapsed mammary tumor had developed continuously from right M3 to M5 (previous location), presenting as a 3 cm diameter cord (13.5 cm long), and extended to the left mammary chain on M4–M5 (4.8 cm long; Figure 1). The right inguinal lymph node was 3 × 4 cm large, while the left one had a normal size. No other abnormal relevant points were noted on clinical examination.

### 3.2. Additional Medical Tests

A tomodensitometric examination (whole body) was performed under anesthesia (including premedication with 0.2 mg/kg methadone and 5 μg/kg medetomidine and induction with titrated propofol, followed by isoflurane gas). It revealed the probable infiltration of the iliac, sacral, suprasternal and right inguinal lymph nodes (LNs). No metastases were identified in the lungs, abdominal organs or brain (Figure 2).

Fine-needle aspirations of the mammary tumor and the right inguinal LN were performed without aspirating, using an air-filled 5 mL syringe and 23-gauge blue needles. The smears were obtained by spreading the samples between two perpendicular glass slides through capillary action and then simply pushing the second slide. The cells were free and round or, more rarely, in clusters, which confirmed the epithelial nature of the proliferation. The cells were medium to large in size with often vacuolated basophilic cytoplasm and had round nuclei, with a high nucleocytoplasmic ratio, and a coarse, reticulated chromatin pattern enriched with several nucleoli. Anisocytosis and anisokaryosis were very significant. Frequent mitosis and some plurinucleate cells were observed, and the right inguinal lymph node was completely infiltrated by the same cells (Figure 3).

An anaplastic carcinoma was suspected. The biochemistry analyses were within normal range (however, ALT: 157 UI/L (standard value < 100 and Alb/Glob ratio of 0.7)). The dog was administered firocoxib (3.56 mg/kg/d). An informed consent form for surgery, stating the extremely unfavorable prognosis, was given to the owner, who still requested surgery. The blood cell count and clotting time were within normal range.

### 3.3. Surgery

Six days after the CT examination, the surgery firstly consisted of ovariectomy with the removal of the four iliac and two sacral LNs, the large removal of the right mammary cord with the right inguinal LN and M2–M5 left mammectomy (Figure 4).

The following day, a sternotomy was performed to remove the suprasternal LN. For this procedure, pain was managed with an epidural injection of morphine 2 h before the surgery and an IV bolus of fentanyl (5 mg/kg) and ketamine (0.6 mg/kg) before induction, which was induced with propofol (2 mg/kg) and diazepam (0.25 mg/kg). Surgery was performed with a continuous rate infusion (CRI) of ketamine (0.6 mg/kg/h) and fentanyl (5 mg/kg/h). The CRI was gradually reduced after surgery within the next 16 h. The dog was released on amoxicillin–clavulanic acid at 12.5 mg/kg BID for 5 days, firocoxib at 3.56 mg/kg once a day and furosemide at 1 mg/kg/d 5 d (to reduce pelvic limb edema). A chloraminophene (2 mg every other day, 6.9 mg/m^2^ EOD) treatment was initiated 10 days after the surgery (Table 1). The wound dehisced in the right distal part, and a second surgery 11 days after the initial surgery was performed to close the wound.

### 3.4. Histological Description

The histopathological examination indicated a grade III solid mammary carcinoma evolving towards prominent lymphatic intravascular carcinomatosis with numerous neoplastic emboli and multifocal LN metastatic extension (all the lymph nodes removed were infiltrated, including the suprasternal LN) (Figure 5), corresponding to stage IV in the WHO classification (T3N2M0). The margins were free of tumoral tissue.

Immunohistochemistry was performed on the tumor and on the infiltrated lymph nodes. The tumor was triple-negative and non-basal-like (ER-negative tumor and 1/1000 ER-positive cells in the LN; PR- and HER2-negative in all localizations, with negative cytokeratin CK5/6-positive cells < 1%).

### 3.5. Adjuvant Treatment

Fourteen days after the initial surgery, the wound healing was good and carboplatin chemotherapy (270 mg/m^2^) was initiated, with maropitant (1 mg/kg/d) being prescribed for 4 days after each chemotherapy treatment; the chloraminophene treatment was continued every other day and firocoxib was administered every day, although a suspension of the latter was prescribed for 7 days after the carboplatin chemotherapy to avoid the risk of inducing renal failure (owner misunderstanding).

A second carboplatin chemotherapy treatment was performed 3 weeks later (202 mg/m^2^ due to neutropenia; Table 1) with the suspension of firocoxib for 13 days, which was reinstated after the occurrence of vulvar edema. Three weeks after the second chemotherapy treatment, the prescapular lymph nodes were enlarged (L: 2 cm; R: 3.5 cm); fine-needle aspiration cytology showed infiltration of these LNs by the same mammary tumor cells. The dog was administered toceranib (2.03 mg/kg EOD) and, 4 days later, the two LNs were removed. The histological examination confirmed their infiltration by tumor cells. The next carboplatin chemotherapy treatment was postponed to 1.5 months after the second one to avoid wound dehiscence and the next chemotherapy treatments (six in total) were performed every 4 weeks (see Table 1). For each chemotherapy treatment, chloraminophene was stopped 2 days before and restored 3 days after the treatment, firocoxib was stopped the day before and restored 8 days later, and the toceranib was stopped the day before and restored 5 days later. Toceranib and chloraminophene were given alternately. Carboplatin doses were gradually reduced to 150 mg/m^2^ due to neutropenia on the day of chemotherapy. After the sixth carboplatin dose, only firocoxib, chloraminophene and toceranib were maintained.

The adverse effects were mostly low-grade (seven events of grade 1 neutropenia and one of grade 3 at the nadir of carboplatine; grade 2 thrombocytopenia at the nadir of the first 270 mg/m^2^ dose of carboplatin and after the sixth session of carboplatin (three consecutive formula counts with grade 3 thrombocytopenia)). This thrombocytopenia was probably due to the evolution of the cancerous disease rather than linked to a direct adverse effect of the treatment. No anemia or gastrointestinal side effect occurred, and ALT and creatinemia always remained within the norms. The hematological and biochemical results at the nadir of chemotherapy at the maximum tolerated dose or on the day of the following chemotherapy treatment are shown in Table 2.

### 3.6. Quality of Life and Follow-Up

The dog had very good quality of life during the whole treatment period, with no weight loss (rather, weight gain from 4 to 5 kg), stable appetite, normal activity and interaction with her owners, with normal habits and usual play phases. After the prescapular LN invasion, she did not show other relapses. After the sixth chemotherapy treatment, she suddenly developed a ventral hematoma with inguinal lymph node reaction. Firocoxib chloraminophene and toceranib treatments were stopped by the veterinary practitioner. Five days later, when they came back for a referral follow-up, the hematoma had shrunk. There was no sign of local recurrence of the mammary tumor, other than an enlarged inguinal LN. The fine-needle aspiration of this LN contained only inflammatory cells and did not indicate tumoral infiltration. The clotting time was within normal values. The blood cell count showed thrombocytopenia, confirmed with a blood smear examination. An ultrasound abdominal examination did not show any abnormal LNs. The dog was administered amoxicillin–clavulanic acid for 10 days and the hematoma resolved.

Five weeks after the last carboplatin chemotherapy treatment, firocoxib and toceranib were reinstated despite persistent thrombocytopenia. The presence of a flaccid tail was noted 7 days later, followed by the acute development of posterior paresis within 24 h. A CT examination (Figure 6) revealed several spinal metastases but no other locations of metastases, and the dog died on the next day. In order to respect the owners’ wishes, necropsy was not performed. The time from initial surgery (carried out in our referral center) to death from the mammary tumor was 218 days (7 months and 6 days).

## 4. Discussion

In the new histologic classification proposed by Goldschmidt in 2011 [10], solid carcinomas are considered highly malignant canine mammary carcinomas. They share the poorest prognosis with anaplastic carcinomas, comedocarcinomas, and “carcinoma and malignant myoepitheliomas” if inflammatory carcinomas are excluded [11,12,13]. In the reported case, the tumor was larger than 5 cm; grew very fast, with poor cellular differentiation; invaded lymphatic vessels, with severe carcinomatosis; and infiltrated sacral, iliac and suprasternal LNs. All of these elements were very-poor-prognosis parameters [11,14]. Histologic stage IIIb has a poor prognosis, with a median survival time of 163 days after surgery [15,16]. Moreover, with numerous infiltrated lymph nodes, without other tissue infiltration demonstrated by tomodensitometric examination, the stage of the reported tumor was at least IV, which further worsened the prognosis [12]. As mammary carcinoma cells tend to invade bone marrow [17,18,19,20], we cannot be certain that bone marrow metastases were not already present, but they were not visible upon the first tomodensitometric examination as bone lesions. We could have carried out a bone marrow puncture and used the cell block technique or a biopsy to look for the presence of epithelial cells in the marrow by immunohistochemistry examination [21], but the round cellular appearance of the tumor cells and their CK5/6 negativity would probably have produced a false-negative result in that case. A cell-free DNA test could have detected the mammary tumor cells, but the detection rate in blood is only about 61% and that in bone marrow is unknown [22].

Very few published case series reported an effect of maximum-tolerated-dose chemotherapy alone on specific survival in dogs bearing canine mammary tumors [1,2,3,4,5,6,7,23,24]. Carboplatin alone [3], in addition to naltrexone [7] or with concurrent metronomic cyclophosphamide chemotherapy [24], seemed to have an interesting effect on survival in female dogs with mammary carcinomas. COX2 overexpression promotes angiogenesis and invasion [25]; the use of anti-COX2 anti-inflammatory drugs has previously shown an effect on certain types of mammary carcinoma, namely, inflammatory mammary carcinomas [23,26,27,28,29]. It seems that firocoxib could also promote the apoptosis of mammary cancer cells, particularly in the case of triple-negative tumors, in vitro and in vivo [30]. In Lavalle’s study, carboplatin was first used alone, without anti-inflammatory drugs, which were given after the last chemotherapy treatment for 6 months [3]. Previously published studies indicated that toceranib had effects on inflammatory mammary carcinomas [28,31], which also invade lymphatic vessels, as did the solid carcinoma described in our case.

In the described case, the owner still requested treatment in spite of the extremely unfavorable prognosis. Given such an aggressive mammary carcinoma, growing very quickly, the treatment had to be intensive to hopefully have an effect. In multi-drug chemotherapy, “the combined agents should possess a demonstrable individual activity against the tumor type being treated, with the potential for synergistic and/or additive activity, as well as disparate mechanisms of action, and as a result distinct adverse event (AE) profiles. Moreover, the agents should ideally be able to be administered concurrently at an optimized therapeutic dose and interval” [32]. The proposed treatment consisted of surgery (comprising radical mastectomy, ovariectomy and the removal of all infiltrated lymph nodes, including suprasternal ones) and combined maximum-tolerated-dose and metronomic chemotherapy treatments with carboplatin, firocoxib, chloraminophene and toceranib. To the best of our knowledge, metronomic chemotherapy and toceranib were previously used together after completing carboplatin sessions and, not at the same time, to treat osteosarcoma [33], and metronomic chemotherapy and piroxicam or firocoxib were administered after completing carboplatin sessions [3]. In a study by London, which included toceranib (2.75 mg/kg EOD), piroxicam (0.3 mg/kg) and cyclophosphamide (10 mg/m^2^ EOD), greater toxicity was described than with piroxicam (0.3 mg/kg) and cyclophosphamide (10 mg/m^2^ EOD) alone [33]. The combination of cyclophosphamide, COX2 inhibitors and toceranib at lower doses has been demonstrated to be well tolerated in dogs bearing inflammatory carcinomas [31]. Toceranib phosphate was administered in the latter study at doses from 2.4 to 2.7 mg/kg PO on a Monday–Wednesday–Friday schedule, and cyclophosphamide was given at a dose of 12.5 mg/m^2^/d PO. In another study, which included piroxicam at 0.3 mg/kg and toceranib at 3.25 mg/kg EOD, 12% of tumor-bearing dogs developed azotemia (where one case was grade IV) and few developed gastrointestinal adverse effects [34]. In our study, the treatment was progressively implemented with the introduction of the different chemotherapeutic compounds in stages. The dose of carboplatin was adapted and progressively reduced according to the neutropenia and thrombocytopenia at the nadir and on the day of each chemotherapy. We selected a lower dose of toceranib compared with the doses described above in the other studies for three reasons: its frequent side effects at the marketing authorization doses [33] or slightly lowered doses [34]; the risk of cumulative effects on liver or kidney function with firocoxib and carboplatin; and, lastly, the fact that using a 15 mg tablet would have led to a large overdose (3.75 mg/kg). As described in a previous study, where female dogs bearing mammary carcinoma were treated with carprofen for 3 months without any effect on renal function [35], no azotemia nor proteinuria occurred with firocoxib, in spite of its association with toceranib and carboplatin. No hepatitis, anemia, diarrhea nor vomiting occurred in our case either, in spite of the concomitant use of carboplatin, toceranib and firocoxib. No thrombocytopenia was noted during treatment (except after the first dose of carboplatine and at the end). Such thrombocytopenia has been previously described [24] but as having occurred cumulatively during the carboplatin and metronomic cyclophosphamide regimen. In Machado’s study [24], the dose of carboplatin used was 300 mg/m^2^ and was not changed in spite of progressive thrombocytopenia during the treatment. In published trials with ITK and concomitant maximal-tolerated-dose chemotherapy, it has been shown that chemotherapy doses had to be reduced. For instance, in a study combining toceranib and carboplatin in tumor-bearing dogs, carboplatin was used at the dose of 200 mg/m^2^ and toceranib at 2.75 mg/kg EOD. No unique or novel adverse events were observed. A dose of carboplatin of 200 mg/m^2^ induced neutropenia (grade I), anemia (grade I) and thrombocytopenia (grade I–III), as well as other events of low grade (gastrointestinal events, hypertension, etc.). Escalating carboplatin doses to 250 or 225 mg/m^2^ increased myelosuppressive effects [36]. The thrombocytopenia occurred around the sixth dose of carboplatin and caused the hematoma observed a few days earlier. It could have occurred due to the bone metastatic process; the cumulative/amplifying effects of the myelosuppression-associated drugs carboplatin, chloraminophene and toceranib; or a dysimmune paraneoplastic mechanism. It did not resolve with chloraminophene and toceranib discontinuation and shortly preceded the discovery of spinal metastases. Paraneoplastic thrombocytopenia was more likely due to either a dysimmune mechanism or the invasion of bone marrow by metastatic carcinoma cells (as shown by the tomodensitometric examination) or both; as we did not carry out a bone marrow examination at the time, we cannot conclude as to the cause of the thrombocytopenia.

The initial tumor was further tested after the dog’s death with immunohistochemistry and was subsequently found to be negative for C-kit and COX2. We could not test for PDGF-R receptors due to a lack of antibodies against PDGR-R; firocoxib and toceranib may have had indirect effects on the tumor microenvironment or through PDGF-R.

Even if we have no proof that the chemotherapy attempted really prolonged this dog’s life, at the very least, we were able to observe that this treatment gave her very good quality of life and did not result in any serious side effects, apart from grade 3 nonclinical neutropenia. Her quality of life was closely monitored based on a sustained and constant dialog with the owner [37] in order to protect the dog from over-treatment, with the owner having been actively alerted with regard to this point at the start of the treatment and throughout.

## 5. Conclusions

A female dog suffering from a severely aggressive mammary carcinoma associated with a very poor prognosis was treated with radical surgery and multimodal therapy combining chemotherapy at the maximum tolerated dose and metronomic therapy, firocoxib and toceranib. To the best of our knowledge, such a treatment has never been published. Adapting the doses of carboplatin, toceranib, firocoxib and chloraminophene prevented the adverse events of the adjuvant treatments and the associated poor quality of life. The dog had a normal life, with no loss of activity, and did continue to interact with her owners, even more than before. The treatment, including the surgeries and the chemotherapy sessions, resulted in 218 days of survival. However, this is only one case; only a prospective study in a cohort of similar cases could determine whether this treatment could prolong the median specific survival time compared with a control population, whether it is safe and whether it could be applied in highly aggressive canine mammary carcinomas.

## Figures and Tables

**Figure 1 animals-14-02618-f001:**
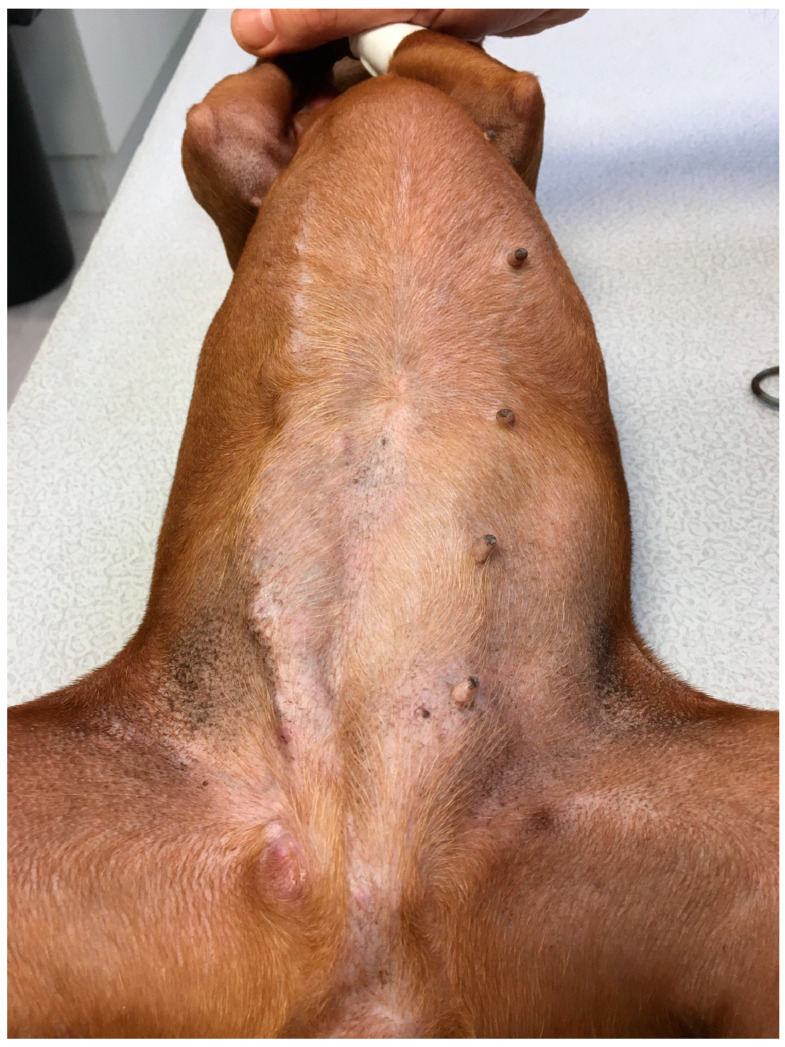
Aspect of right mammary scar on day of referral consultation.

**Figure 2 animals-14-02618-f002:**
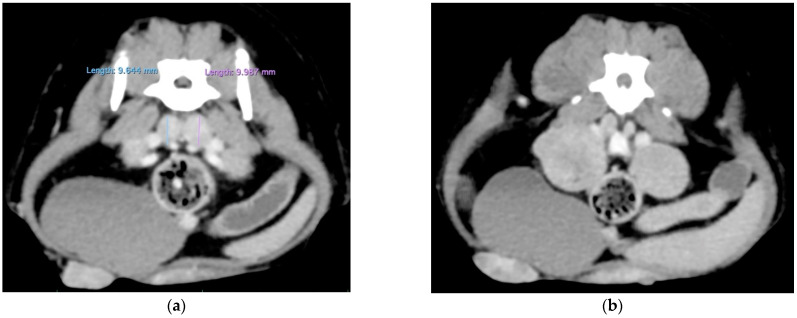
Tomodensitometric examination before surgery. The right inguinal, iliac medial (**a**), lateral (**b**), sacral (**c**) and suprasternal (**d**) lymph nodes were increased in size and rounded in shape, with a strong homogeneous enhancement following the injection of the contrast medium and were strongly suspected to be infiltrated. No metastases were identified in the lungs, abdominal organs or brain.

**Figure 3 animals-14-02618-f003:**
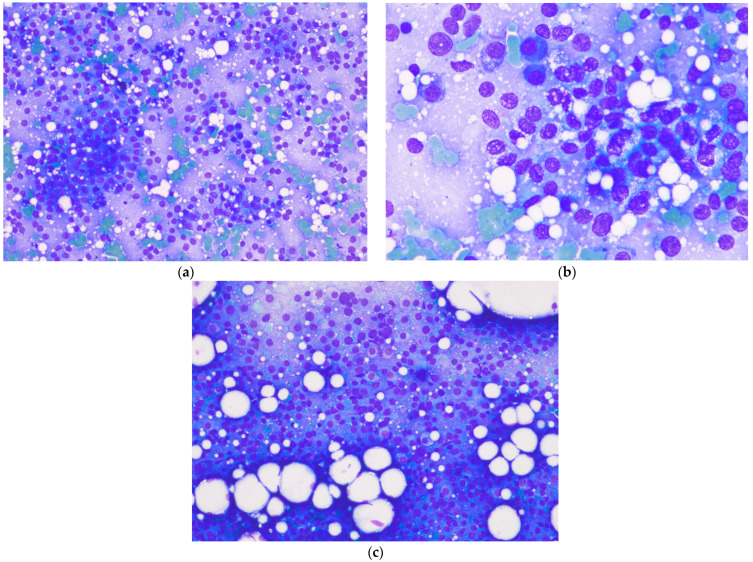
Fine-needle aspiration of mammary tumor and right inguinal LN. May Grunwald Giemsa staining. (**a**) Magnification: ×10. The cells were free and round or, more rarely, in clusters, which confirmed the epithelial nature of the proliferation. (**b**) Magnification: ×100. The cells were medium to large in size with often vacuolated basophilic cytoplasm and had round nuclei, with a high nucleocytoplasmic ratio, and a coarse, reticulated chromatin pattern enriched with several nucleoli. Anisocytosis and anisokaryosis were very significant. Frequent mitosis and some plurinucleate cells were observed. (**c**) Magnification: ×10. The inguinal lymph node was completely infiltrated by the same cells. An anaplastic carcinoma was suspected. Photography: Delphine Rivière; @inovievet, Montpellier, France.

**Figure 4 animals-14-02618-f004:**
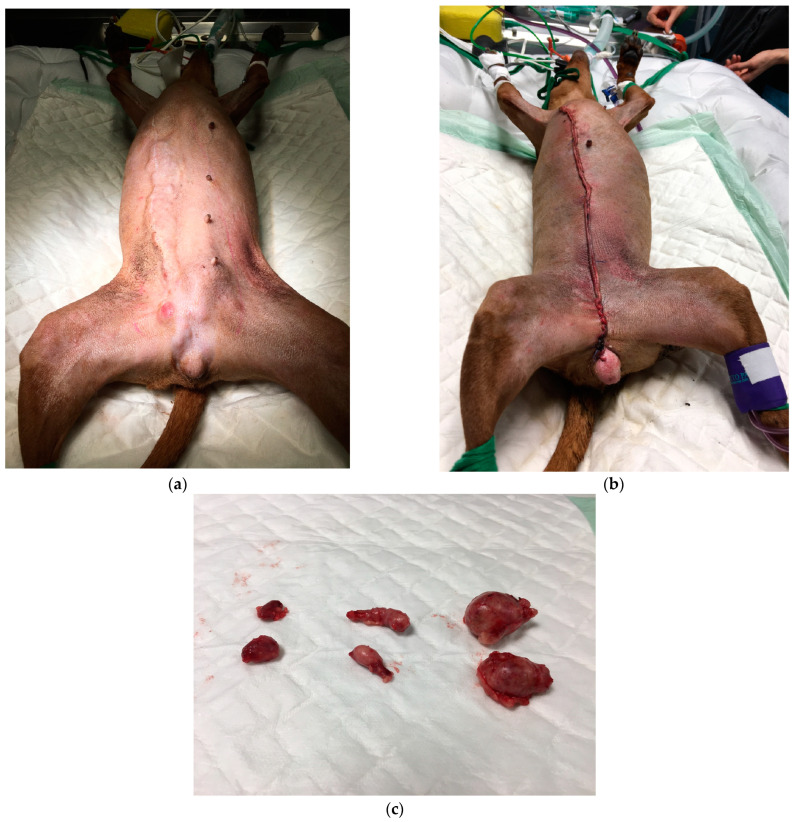
(**a**) Aspect of mammary tumor relapse on day of surgery. The tumor had further grown within 6 days (see, for comparison, Figure 1). An ovariectomy was performed first, with the removal of iliac and sacral lymph nodes, followed by a complete radical excision of the right relapsing cord, which had grown fast in a week. The surgical plan was drawn with a red pencil. (**b**) The scar after the surgery. (**c**) The removed sacral and iliac lymph nodes.

**Figure 5 animals-14-02618-f005:**
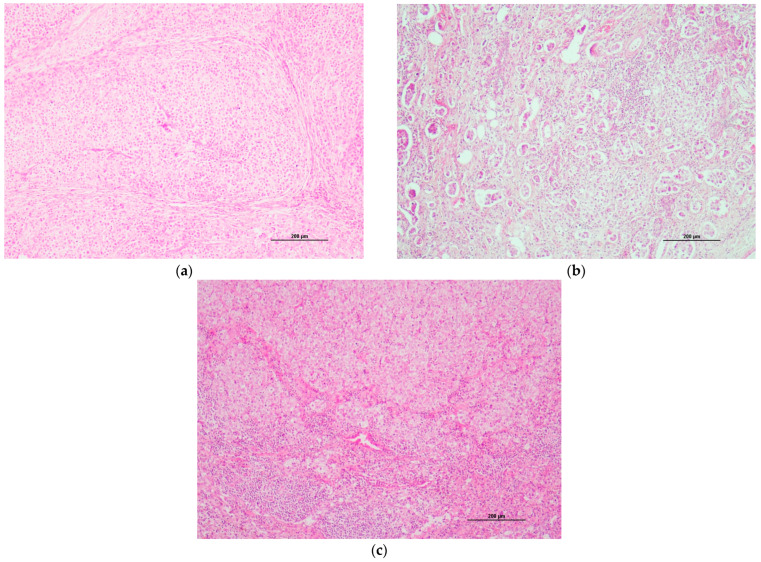
(**a**) Histopathological examination of the mammary tumor and lymph nodes. Hematoxylin–eosin–saffron (HES) staining. The mammary tumor was invasive ((**a**) ×10) with lymphatic intravascular invasion ((**b**) ×20). The examination indicated a grade III solid mammary carcinoma. Multifocal LN metastatic extension was noted ((**c**) ×10). Photography: @Oniris, Nantes, France.

**Figure 6 animals-14-02618-f006:**
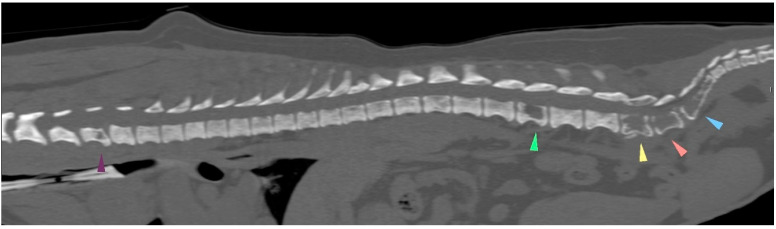
The CT scan examination at the end. The C6, L3, L6, L7, S1 and S2 vertebral bodies showed cookie-cutter lesions, pointed by the arrows. (The lesion at L6 severely invaded the medullary canal; not shown here).

**Table 1 animals-14-02618-t001:** Treatment plan, weight (kg) and clinical events.

Date	Weight	Surgery	Molecular Compounds						Clinical Events
			Carboplatine	Chloraminophene		Firocoxib, 14.2 mg/d		Toceranib, 10 mg	
27 January 2021-28 j0	4								
28 January 2021									
7 February 2021 j+11				2 mg EOD					
6 February 2021									wound dehiscence
8 February 2021		wound dehiscence surgery							
10 February 2021									
11 February 2021 j+25	4		270 mg/m²						
18 February 2021				2 mg EOD					
4 March 2021									
5 March 2021 j+37	4.1		202 mg/m²						
13 March 2021				2 mg EOD					
18 March 2021	4.450								vulvar edema,
26 March 2021	4.1							2.43 mg/kg EOD	vulvar edema, granulation of mammary tissue (ex Left M4–M5 location), bilateral increased prescapular LN.
30 March 2021 j+62		prescapular lymph nodes							
16 April 2021	4								correct surgical scar
18 April 2021									
20 April 2021									
21 April 2021									
22 April 2021									
23 April 2021 j+86	4.2		162 mg/m²						
26 April 2021				2 mg every 3 days					
28 April 2021								2.43 mg/kg EOD	
1 May 2021									
10 May 2021	5								normal clinical examination
18 May 2021									
19 May 2021									
20 May 2021 j+113	4.89		151 mg/m²						
29 May 2021				2 mg every 3 days					
1 June 2021									
3 June 2021									
12 June 2021									
13 June 2021									
22 June 2021									
23 June 2021									
25 June 2021 j+149	4.850		151 mg/m²						normal clinical examination
29 June 2021				2 mg every 3 days					
1 July 2021									
3 July 2021									
10 July 2021	5.250								
21 July 2021									
22 July 2021									
23 July 2021 j+175	4.930		151 mg/m²						normal clinical examination
27 July 2021				2 mg every 3 days					
29 July 2021									
31 July 2021									
2 August 2021 j+187									left himb limb edema and ventral hematoma
7 August 2021									left pelvic member edema, ventral hematoma, inguinal LN 18 mm (not infiltrated by cancer cells)
16 August 2021									the hematoma is shrinking and the limb edema has disappeared
25 August 2021									
1 September 2021									flaccid tail, evolving into posterior paresis within 24 h, spinal metastasis (CT examination)
2 September 2021 j+218									death

**Table 2 animals-14-02618-t002:** Serial blood cell counts and biochemistry results. SV, interval of standard values; PCV, packed cell volume; RBCs, red blood cells; WBCs, white blood cells; ALT, alanine amino transferase. Values not included in the reference intervals are highlighted in yellow.

Date	RBCs (10^12^/L) SV [5.6–8.8]	PCV (%) SV [37.3–61.7]	Hb (g/dL) SV [13.1–20.5]	WBCs (10^9^/L) SV [5.05–16.7]	Neutrophils (10^9^/L) SV [2.9–11.6]	Monocytes (10^9^/L) SV [0.16–1.12]	Lymphocytes (10^9^/L) SV [1–5.1]	Platelets (10^3^/µL) SV [148–484]	ALT (IU/L) SV [10–125]	Blood Creatinine (mg/L) SV [5–18]
27–28 January 2021	7.85	49.4	17.3	9.33	7.83	0.61	0.68	195	157	5
5 March 2021	7.27	46.8	18.3	3.6	1.85	0.14	1.62	198	160	5
18 March 2021	7.36	46.5	15.9	6.16	2.94	0.36	2.85	65		
26 March 2021	7.09	44.9	15.4	6.77	4.41	0.23	2.12	175	88	6.4
23 April 2021	6.92	44	15.1	4.69	3.94	0.3	0.19	314	56	5.8
10 May 2021	6.97	45.1	18.8	3.06	1.93	0.08	1.05	220	41	10.5
20 May 2021	7.75	50.3	16.8	4.18	3.52	0.27	0.17	266		
5 June 2021	7.61	49	17	2.75	1.61	0.06	1.08	145	32	7.3
16 June 2021	8.05	52.5	15.4	4.18	2.61	0.06	1.51	158		
25 June 2021	7.06	45.9	15.4	3.55	2.67	0.32	0.13	176	41	5.4
10 July 2021	6.88	45.5	15.7	2.88	2.2	0.07	0.61	147	32	10
7 August 2021	5.83	37.5	12.9	1.31	0.88	0.11	0.09	39		
16 August 2021	6.94	44.9	14.6	3.83	3.32	0.08	0.42	35		
25 August 2021	7.24	47.7	16	4.66	3.63	0.15	0.88	42		

## Data Availability

The original contributions presented in the study are included in the article, further inquiries can be directed to the corresponding author.

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
