# Peer review of "Intensive Multimodal Chemotherapy in a Dog Suffering from Grade III/Stage IV Solid Mammary Carcinoma"

_animals, 2024, doi:10.3390/ani14172618_

Round 1

Reviewer 1 Report

Comments and Suggestions for Authors

In this case report the authors report a canine high grade/high stage mammary tumor that was treated with a multi modal approach and reached a fairly long OS with acceptable tox.

The paper has its merits mostly on reporting potential benefits with adjuvant chemo post op. However, in the present form it is very hard to follow.

I recommend the authors to send the manuscript for English proof-reading. The text is also too wordy in the present version and can be significantly shortened.

The table 1 is currently very hard to read. If this is due to the odd formatting in the manuscript, or if the authors have been asked to present the case- accordingly is hard to know. Please format the table 1 correctly before re-submission.

It is currently impossible to call this a "protocol", as the dose adjustments are very frequent in the text. If supposed to be a "protocol", please provide a table summarizing the treatment protocol.

AE grading should also be mentioned in the text in the manuscript and not only in table 1. Currently Table 1 is very hard to read and hence tox is hard to appreciate.

I suggest to use chlorambucil instead of chloraminophene.

Figures are relevant and good.

Comments on the Quality of English Language

Even if not native English, I do not think the language is of satisfactory quality. The writing is also often too complicated and "wordy" and could be shortened without loosing in quality. Please send for proof-reading.

Author Response

Dear reviewer (1)

Thank you for your help to improve the manuscript, we hope to have responded to all your comments according to your expectations.

Here are our detailed responses in your text.

Comments and Suggestions for Authors

In this case report the authors report a canine high grade/high stage mammary tumor that was treated with a multi modal approach and reached a fairly long OS with acceptable tox.

The paper has its merits mostly on reporting potential benefits with adjuvant chemo post op.. However, in the present form it is very hard to follow.

I recommend the authors to send the manuscript for English proof-reading. The text is also too wordy in the present version and can be significantly shortened. We have had the text corrected after answering your questions and those of the second reviewer by the MDPI editing services.

The table 1 is currently very hard to read. If this is due to the odd formatting in the manuscript, or if the authors have been asked to present the case- accordingly is hard to know. Please format the table 1 correctly before re-submission. It is currently impossible to call this a "protocol", as the dose adjustments are very frequent in the text. If supposed to be a "protocol", please provide a table summarizing the treatment protocol. We agree that Table I was hard to read. It has been splitted in two Tables and amended as requested to summarize the treatment plan. The term ‘protocol’ is indeed a misnomer; the term ‘treatment plan’ seems more appropriate and has been applied.

AE grading should also be mentioned in the text in the manuscript and not only in table 1. Currently Table 1 is very hard to read and hence tox is hard to appreciate.  We added a paragraph in the text regarding side effects, and we also created a table II to show the hematological and biochemical side effects of the treatment.

I suggest to use chlorambucil instead of chloraminophene. We did not use chlorambucil, but the generic version, hence the choice of the molecule's name.

Figures are relevant and good.

We just want to add that the announced survival time was incorrectly calculated at the start and that after the renewal of the calculation it was 218 days and not 237 days (our apologies for the error).

Comments on the Quality of English Language

Even if not native English, I do not think the language is of satisfactory quality. The writing is also often too complicated and "wordy" and could be shortened without loosing in quality. Please send for proof-reading. We have submitted it to MDPI's publishing department.

Best regards,

Claire Beaudu-Lange and Emmanuel Lange

Reviewer 2 Report

Comments and Suggestions for Authors

The manuscript provides a single case description of the use of surgery combined with multimodal chemotherapy in State IV canine mammary carcinoma.  Though the authors describe only a single case, the information will be of interest to readers due to the relative dearth of similar information.  I ask the authors to consider the following comments:

1. In the title, change to "...Chemotherapy in a dog..."

2. The authors refer several times to "high dose surgery."  I am confused by this term and wonder if the authors mean to state, "radical mastectomy."  It may be better to simply remove the adjective "high dose" and simply use the term "surgery."

3. On line 71, please state the scanning modality (tomodensitometry) that was used.  Was the dog sedated for scanning? If so, please state the sedative, dosage, and route.

4. Line 72: provide some descriptive details of the method used for fine-needle aspiration and note the anatomic sites that were sampled.

5. Line 74: provide some brief detail regarding the histological and immunohistochemical methods used, including fixative, thickness of tissue sections, and staining methods. 

6. Line 78:  Were the discussions with the owners structured in any way?  For example, was a list of criteria reviewed each time?  

7. Several times, the authors described procedures (e.g., FNA, line 107) as being "realized."  It may be more appropriate to describe procedures as having been "performed."

8. Figure 3:  indicate the stain used. Also, the size of the cells in 3a look the same as that in 3c, yet the magnification is described in the legend as being 10X for 3a and 100X for 3c.  Please make sure the legend is correct in terms of the magnifications.

9. Line 176 refers to a "short suspension."  Please state specifically for how long treatment was halted.

10. Line 201/202 seems like an incomplete sentence.  Perhaps, the authors mean to say, "...other than an enlarged..." rather than "...but an enlarged..."

11. Lines 208/209: restate as, "Presence of flaccid tail was noted, followed by acute development of posterior paresis within 24 h."

12. Lines 211/212: change "...necropsy was not performed, nor was vertebral histology." to "...necropsy was not performed."   It logically follows that if necropsy was not performed that neither would vertebral histology be performed.

13. Lines 225 and 228.  In line 225 you note that stage IIIb has the "worst" prognosis, but in line 228 you state that the tumor was stage IV which "worsened" the prognosis.  Thus, stage IIIb is not the worst prognosis.  Instead, you may wish to change line 225 to ''...stage IIIb has a poor prognosis..."

14. Line 247: what is meant by "instated?" do you mean that carboplatin treatment was initiated?

15. Line 267: capitalize "cox2" to "COX2"

16. Line 268: change "proven" to "demonstrated."

17. Line 275: change "molecules" to "chemotherapeutic compounds."

18. In the discussion where thrombocytopenia is discussed, please note the likely contribution of thrombocytopenia to hematoma in this dog.

19. Line 31: you describe the dog as having "very good quality of life."  However, that is debatable, given that the dog had to undergo two surgeries, multiple doses of chemotherapy, and experienced a hematoma.  Further, there is the possibility that the owners overstated the quality of life via unrealistic optimism.  It is probably fair to say that the dog had a "reasonably good" quality of life.

Comments on the Quality of English Language

There are some minor language issues as noted in my comments to the authors.  The manuscript will need to undergo a small amount of editing to ensure use of standard English.

Author Response

Dear reviewer (2),

Thank you for your help to improve the manuscript, we hope to have responded to all your comments according to your expectations.

Here are our detailed responses in your text.

The manuscript provides a single case description of the use of surgery combined with multimodal chemotherapy in State IV canine mammary carcinoma.  Though the authors describe only a single case, the information will be of interest to readers due to the relative dearth of similar information.  I ask the authors to consider the following comments:

  1. In the title, change to "...Chemotherapy in a dog..." done
  2. The authors refer several times to "high dose surgery."  I am confused by this term and wonder if the authors mean to state, "radical mastectomy."  It may be better to simply remove the adjective "high dose" and simply use the term "surgery." The surgery was described as a « high dose surgery »,  as we had performed a radical mastectomy, ovariectomy, removal of the pelvic and iliac NL and removal of the susternal NL,which is rarely seen in canine mammary surgeries. We used the term surgery as recommended.
  3. On line 71, please state the scanning modality (tomodensitometry) that was used.  Was the dog sedated for scanning? Yes she was.  If so, please state the sedative, dosage, and route. Done, we added the sedation modality.
  4. Line 72: provide some descriptive details of the method used for fine-needle aspiration and note the anatomic sites that were sampled. Supplementary details were added as per your request.
  5. Line 74: provide some brief detail regarding the histological and immunohistochemical methods used, including fixative, thickness of tissue sections, and staining methodsWe added these methods on the material and methods.
  6. Line 78:  Were the discussions with the owners structured in any way?  For example, was a list of criteria reviewed each time?  Thank you for the question, we answered in the text in the section Material and Methods.
  7. Several times, the authors described procedures (e.g., FNA, line 107) as being "realized."  It may be more appropriate to describe procedures as having been "performed." We made those corrections.
  8. Figure 3:  indicate the stain used. Also, the size of the cells in 3a look the same as that in 3c, yet the magnification is described in the legend as being 10X for 3a and 100X for 3c.  Please make sure the legend is correct in terms of the magnifications. The legend was indeed incorrect, it has been edited.
  9. Line 176 refers to a "short suspension."  Please state specifically for how long treatment was halted. We stated it in the text and modified the Table I, splitting it in two Tables, so that they could be easier to read.
  10. Line 201/202 seems like an incomplete sentence.  Perhaps, the authors mean to say, "...other than an enlarged..." rather than "...but an enlarged..." Correction made.
  11. Lines 208/209: restate as, "Presence of flaccid tail was noted, followed by acute development of posterior paresis within 24 h." Correction made
  12. Lines 211/212: change "...necropsy was not performed, nor was vertebral histology." to "...necropsy was not performed."   It logically follows that if necropsy was not performed that neither would vertebral histology be performed. Correction made
  13. Lines 225 and 228.  In line 225 you note that stage IIIb has the "worst" prognosis, but in line 228 you state that the tumor was stage IV which "worsened" the prognosis.  Thus, stage IIIb is not the worst prognosis.  Instead, you may wish to change line 225 to ''...stage IIIb has a poor prognosis..." Correction made
  14. Line 247: what is meant by "instated?" do you mean that carboplatin treatment was initiated? Carboplatin had been initiated alone, and the dogs were treated with anti-inflammatory drugs only after the end of the 4 carbo-chemotherapies, correction made.
  15. Line 267: capitalize "cox2" to "COX2" corrections made
  16. Line 268: change "proven" to "demonstrated." corrections made
  17. Line 275: change "molecules" to "chemotherapeutic compounds." corrections made
  18. In the discussion where thrombocytopenia is discussed, please note the likely contribution of thrombocytopenia to hematoma in this dog. Correction made
  19. Line 31: you describe the dog as having "very good quality of life."  However, that is debatable, given that the dog had to undergo two surgeries, multiple doses of chemotherapy, and experienced a hematoma.  Further, there is the possibility that the owners overstated the quality of life via unrealistic optimism.  It is probably fair to say that the dog had a "reasonably good" quality of life. Indeed both surgeries were aggressive, however the dog recovered extremely quickly with our careful pain management; Throughout the treatment, she was extremely happy, even happy to come to the clinic, and continued her life without any changes described by her owners. The edema of the pelvic limb was temporary (at the beginning and at the end), and did not seem to make her suffer (she did not limp, and did not lick herself, and went on walking). We do not think that this is only linked to exaggerated optimism on the part of the owner, or on our part as caregivers, the entire medical team at the clinic having made this observation: We were able to see for ourselves her joy of living. Our team, nurses and vets, is very sensitive to the quality of life of animals subjected to anticancer treatments, and pay real attention to it (the last publication cited, unfortunately in French, corresponds to work done on this subject by us in collaboration with another oncologist and a veterinary behavior specialist). It seems right to us in this case to describe a certainly unexpected but very good quality of life.

We just want to add that the announced survival time was incorrectly calculated at the start and that after the renewal of the calculation it was 218 days and not 237 days (our apologies for the error).

Comments on the Quality of English Language

There are some minor language issues as noted in my comments to the authors.  The manuscript will need to undergo a small amount of editing to ensure use of standard English.We have submitted it to MDPI's publishing department.

Best regards,

Claire Beaudu-Lange and Emmanuel Lange

Round 2

Reviewer 1 Report

Comments and Suggestions for Authors

Thank you for the updated manuscript. Language improved. Good to split table 1 into two and they are now easier to follow. Also good that how the assessment of QoL was done now is described in the text in more details. I still think that the manuscript is too wordy (too long) for a case report, but I leave this to the Editor to decide if the amount of text is according to instructions for a case report. 

Otherwise, I have no further comments.